# Realizing Both Antibacterial Activity and Cytocompatibility in Silicocarnotite Bioceramic via Germanium Incorporation

**DOI:** 10.3390/jfb14030154

**Published:** 2023-03-14

**Authors:** Yingqi Ji, Shun Yang, Jian Sun, Congqin Ning

**Affiliations:** 1State Key Laboratory of High Performance Ceramics and Superfine Microstructure, Shanghai Institute of Ceramics, Chinese Academy of Sciences, Shanghai 200050, China; 2Center of Materials Science and Optoelectronics Engineering, University of Chinese Academy of Sciences, Beijing 100049, China; 3Department of Prosthodontics, Shanghai Ninth People’s Hospital, Shanghai Jiao Tong University School of Medicine, Shanghai 200011, China; 4The Education Ministry Key Lab of Resource Chemistry, Shanghai Key Laboratory of Rare Earth Functional Materials and Shanghai Frontiers Science Center of Biomimetic Catalysis, Shanghai Normal University, Shanghai 200234, China

**Keywords:** implant material, silicocarnotite, germanium dioxide, antibacterial activity, cytocompatibility

## Abstract

The treatment of infective or potentially infectious bone defects is a critical problem in the orthopedic clinic. Since bacterial activity and cytocompatibility are always contrary factors, it is hard to have them both in one material. The development of bioactive materials with a good bacterial character and without sacrificing biocompatibility and osteogenic activity, is an interesting and valuable research topic. In the present work, the antimicrobial characteristic of germanium, GeO_2_ was used to enhance the antibacterial properties of silicocarnotite (Ca_5_(PO_4_)_2_SiO_4_, CPS). In addition, its cytocompatibility was also investigated. The results demonstrated that Ge–CPS can effectively inhibit the proliferation of both *Escherichia coli* (*E. coli*) and *Staphylococcus aureus* (*S. aureus*), and it showed no cytotoxicity to rat bone marrow-derived mesenchymal stem cells (rBMSCs). In addition, as the bioceramic degraded, a sustainable release of germanium could be achieved, ensuring long-term antibacterial activity. The results indicated that Ge–CPS has excellent antibacterial activity compared with pure CPS, while no obvious cytotoxicity was observed, which could make it a promising candidate for the bone repair of infected bone defects.

## 1. Introduction

With rapid increases in bone cancer, trauma, plastic surgery, joint and spinal diseases and osteoporotic diseases, the demand for medical devices and implants in orthopedics has grown explosively over the past few decades [1,2]. Motivated by clinical demands, the development of hard tissue implant materials has mainly concentrated on improving and optimizing biocompatibility and mechanical properties [3]. Because of the similarity of chemical properties to natural bone minerals, calcium phosphate bioceramics, like β-Tricalcium phosphate (β-TCP), and hydroxyapatite (HA), have been widely used in orthopedic clinics showing good biocompatibility and bone conductivity [4,5,6]. Nevertheless, their ability to induce osteogenic differentiation is restricted [7,8,9]. It has been reported that silicon plays an important role in the early stage of bone formation [10,11], and many studies have further confirmed that the silicon component is very critical in the heterogeneous biological mineralization process of calcium silicate-based composites [12]. Compared with pure HA, silicon-substituted HA has a better bone regeneration performance. What is noteworthy is that silicocarnotite (CPS, Ca_5_(PO_4_)_2_SiO_4_), a silicon-containing calcium phosphate phase, is always present in silicon hydroxyapatite [13,14,15]. Subsequent studies have found that CPS is superior to pure HA in apatite formation, osteogenic activity, and biodegradability [16,17]. In addition, CPS has been shown to promote tendon-bone healing [18,19]. Moreover, it has been found that nano-crystallized silicocarnotite powder prepared by an ultrasound-assisted aqueous precipitation method exhibits enhanced sinterability [20].

It is known that bacterial infections usually happen in surgical sites after orthopedic implantation, which has a negative effect on bone regeneration and integration [21,22]. What is worse, it may even lead to surgical failure [23,24,25]. It is reported that the infection rate of implanting-related cases is 5% of primary cases and 6% of renovation cases, and has risen significantly to 43% of previous infection cases [26]. Therefore, bacterial infection after implantation is still one of the most common and severe medical complications [27]. The conventional way to treat bacterial infections is by using antibiotics [28]. However, the rapid spread and development of antibiotic resistance in microbes, caused by the overuse of antibiotics, have become a main concern and an enormous challenge in treating implant-associated infections [29]. Accordingly, developing anti-infectious implant materials without using antibiotics is a promising way to solve this problem.

Germanium (Ge) was first discovered and named by Winkler in 1886, and it is a trace element in animals and plants. In recent years, it has been found that some medical materials and food containing germanium have particular healthcare functions [30]. Germanium oxide is an inorganic compound that has been confirmed to improve the physical and mechanical properties of glass polyalkenoate cement [31]. It has been reported that germanium has not only antiviral, and anti-inflammatory activities, but also anticancer and anti-tumor functions by scavenging free radicals [32]. Especially, it is worth noting that organic germanium possesses an excellent antibacterial performance [33]. The organic germanium compound, 2-carboxyethyl germanium sesquioxide (Ge-132), can inhibit the propagation of lactic acid bacteria. Katakura, T et al. found that macrophages of thermally injured mice previously treated with Ge-132 protected the mice depleted of neutrophils and macrophages against *methicillin-resistant S. aureus* infection [34]. As a form of inorganic compound germanium, GeO_2_ has shown a similar anti-tumor effect and antibacterial activity as Ge-132 [35]. C. P. Avanzato et al. found that nanocrystalline magnesium oxide–germanium dioxide nanocomposite powders possess a prominent antibacterial effect on *S. aureus* and *E. coli* [36]. Chauhan NPS et al. found that Pectin–GeO_2_ nanocomposite could inhibit the propagation of *S. pyogenus, E. coli*, *S. aureus*, and *P. aeurginosaand* [37].

To gain a desirable orthopedic implant material that can reduce the formation of biofilm or inhibit bacterial attachment and improve ideal tissue responses such as osseointegration, CPS ceramics with different GeO_2_ contents (Ge–CPS) were prepared and the effects of GeO_2_ on the antibacterial activity and cytocompatibility of CPS ceramics were investigated.

## 2. Materials and Methods

### 2.1. Materials

Tetraethyl orthosilicate (TEOS, AR) and ethyl alcohol (C_2_H_5_OH, AR) were purchased from Shanghai Ling Feng Chemical Reagent Co., Ltd. (Shanghai, China). Calcium hydroxide (Ca(OH)_2_, AR) and phosphoric acid (H_3_PO_4_, AR), and germanium dioxide (GeO_2_, 99.999%) were obtained from Sinopharm Chemical Reagent Co., Ltd. (Shanghai, China). Polyethylene glycol (PEG) and polyvinyl alcohol (PVA) were obtained from Aladdin Reagent (Shanghai, China).

### 2.2. Synthesis of Ge–CPS Powders and Bioceramics

The CPS and Ge–CPS powders with designed Ge/(Ge + Si) molar ratios of 0%, 5%, 10%, and 15% were synthesized through an ultrasound-assisted aqueous precipitation method as shown in Figure 1a. The obtained products were labeled as CPS, 0.05Ge–CPS, 0.10Ge–CPS, and 0.15Ge–CPS, respectively. After that, the CPS and Ge–CPS powders were ball-milled with ethyl alcohol and PVA for 2 h. To reduce the influence of porosities on the biological properties of different samples, polyethylene glycol (PEG) particles with a size of 150~425 µm were added as pore formers. Afterwards, the powders were mixed with PEG and pressed into a disc with dimensions of 2 mm × 10 mm. Then the discs were sintered at 1200 °C for 2 h in a muffle furnace to produce porous CPS and Ge–CPS scaffolds, and the porosity of all scaffolds was about 60%.

### 2.3. Characterization of the Ge–CPS Powders and Bioceramics

The phase composition of starting powders and sintered Ge–CPS ceramics were characterized by X-ray diffraction (XRD, D/MAX-RBX, Rigaku, Japan) with Cu Kα radiation. The morphology of the prepared powders and the microstructure of the Ge–CPS bioceramic scaffolds were observed by scanning electron microscopy (SEM, S-3400N Typel, HITACHI, Tokyo, Japan) with an energy-dispersive X-ray spectrometer (EDS). Raman spectra were detected by a Horiba LabRAM HR spectrometer using a laser with an excitation wavelength of 532 nm and 5% of the total laser intensity.

### 2.4. Release of Germanium in Vitro

The release of Ge of different scaffolds was detected by refreshing the entire liquid volume. All the samples and PBS were sterilized. The scaffolds were placed into 10 mL PBS (pH 7.4) and incubated at 37 °C. At each time point, the release of Ge of different samples was determined. Meanwhile, the release medium of samples was completely removed at a predetermined time point and replaced with 10 mL fresh PBS. The Ge concentrations of the specimens were tested by inductively coupled plasma atomic emission spectrometry (ICP-AES; Vista MPX, Varian, Palo Alto, CA, USA).

### 2.5. Bacterial Activity Assays 

#### 2.5.1. Bacteria Strains

*E. coli* and *S. aureus* were cultured to detect the antimicrobial property of CPS and Ge–CPS scaffolds. The *E. coli* and *S. aureus* were cultured by Luria-Bertani (LB) and Trypticase Soy Broth (TSB) medium, respectively. Before culturing, 0.9% NaCl solution which was autoclaved, was used to dilute the bacterial suspension to a suitable concentration. The specimens were placed in 24-well plates and disinfected with 75 vol% alcohol for 2 h and then the 24-well plates with specimens were sterilized with a UV light for 12 h. After that, 1 mL bacterial suspension (10^7^ cfu/mL) was dropped into each well with samples and then cultured in an incubator at 37 °C (all the bacterial experiments were conducted in a general aseptic environment).

#### 2.5.2. Viability of Bacteria

After being cultured for 1 day, 0.5 mL 10% Alamar blue solution (the mixture of Alamar blue and 0.9% NaCl solution) was introduced to each well with samples and incubated for another 2 h. Afterwards, 100 µL reaction solution was transferred to a 96-well plate, and the fluorescence intensity was reduced Alamar blue in the solution was investigated at wavelengths of 560 and 590 nm with an enzyme-labeled instrument. The proliferation of bacteria was calculated under the instruction of the Alamar blue assay. To count the bacterial colony, 10 μL suspensions cultured with samples were carefully diluted 100-fold with 0.9% NaCl solution. After that, 100 μL diluted solution was added to LB or TSB agar culture medium and incubated at 37 °C for another 18 h. The colonies were counted and photographed by a Gel imaging system (FluorChem M, Protein Simple, San Jose, CA, USA).

#### 2.5.3. Morphology of Bacteria

The specimens were washed three times with 0.9% NaCl solution after incubating for 1 day and transferred to a new 24-well plate and then immobilized with 0.8 mL glutaraldehyde solution for 12 h. After that, the samples were dehydrated via a group of ethanol solutions. After anhydration, the specimens were dried with a mixture of hexamethyldisilane (HMDS) and ethanol. Afterwards, the scanning electron microscope (SEM, S3400, HITACHI, Japan) was used for the observation of the bacteria. Before observation, the specimens were prepared by coating them with a thin film of platinum (Pt), which was deposited by sputtering before microscopy.

#### 2.5.4. Bacteria Live/Dead Staining

The Live/Dead BcaLight kit was used to stain the bacteria on the sample surface. In summary, after being cultured for 1 day, the bacteria were washed three times with 0.9% NaCl solution, and then 500 μL diluted dye was introduced to each specimen, staining at 37 ℃ in the oven for 20 min. After that, the specimens were washed twice with 1 mL 0.9% NaCl solution, and then the fluorescence microscope was used to observe the specimens above.

### 2.6. Cytocompatibility Assessment of Ge–CPS Bioceramics

#### 2.6.1. rBMSCs Culture

The rBMSCs were used for investigating the cytocompatibility of the Ge–CPS specimens. The cells were obtained from the Stem Cell Bank, Chinese Academy of Sciences, and were cultured at 37 °C in a 5% CO_2_ incubator with 10 mL of the α-Minimum Essential Medium (α-MEM) replenished with 10% fetal bovine serum (FBS, Gibco, Invitrogen, Inc., Waltham, MA, USA) and 1% antimicrobial which, is a mixture of penicillin and streptomycin (2A, Gibco, Invitrogen Inc.). The medium was changed every 2 days. When the proliferation rate of cell lines reached up to 90%, the cells were detached by a trypsin/EDTA solution, and then were subcultured for biological evaluation (all the cell experiments were conducted in an aseptic environment).

#### 2.6.2. Cell Viability Assay

The sterilized Ge–CPS scaffolds were placed in the wells of 24-well plates separately, and then 1 mL rBMSCs suspension, with a density of 2.5 × 10^4^ cell/mL, was introduced to each well. A solution of α-MEM with 10% FBS and 1% 2A was used to prepare the above cell suspension. Pure CPS scaffolds were used as the control group. After being cultured for 1, 3, 5, and 7 days, the cells on the specimens were washed three times by the phosphate buffer saline (PBS) and incubated with a 0.5 mL mixture of α-MEM and 10% Alamar blue for another 2 h. Afterwards, the fluorescence intensity of Alamar blue was measured with an enzyme-labeled instrument.

#### 2.6.3. Cell Adhesion

A 1 mL rBMSCs cell suspension whose density is 5.0 × 10^4^ cell/mL was inoculated into the disinfected specimens. After 24 h culturing, the samples were washed three times with PBS and then fixed with 4% paraformaldehyde (PFA) solution for 10 min. Then, using the Triton X-100 to permeabilize the cells on the surface of the scaffolds, the cells above were stained by FITC and DAPI. Finally, fluorescence microscopy was used to observe the F-actin and cell nuclei.

### 2.7. Statistical Analysis

The outcome values are summarized in terms of means ± standard deviations. The statistically significant difference (*p*) was calculated by one-way ANOVA with the GraphPad Prism statistical software package. A value of *p* < 0.05 (denoted by *), *p* < 0.01 (denoted by **), and *p* < 0.001 (denoted by ***) were considered significant differences.

## 3. Results

### 3.1. Characterization of the Ge–CPS Powders

The XRD patterns and morphologies of the CPS precursor and the as-purchased GeO_2_ powder are presented in Figure 1b. It shows that the diffraction pattern of the CPS precursor coincides well with the standard PDF card of hydroxyapatite (PDF# 09-0432), and the XRD pattern of the GeO_2_ powder coincides very well with the standard PDF card #83-0543. From the SEM images in Figure 1b, the CPS precursor exhibits a fusiform-like structure and the GeO_2_ powder shows an appearance of micron-size agglomerates. The morphology of the CPS powders is presented in Appendix A.

The XRD patterns of the calcined Ge–CPS powder are displayed in Figure 1c, which reveals that GeO_2_ does not affect the main crystalline phase of CPS which is still Ca_5_(PO_4_)_2_SiO_4_ (PDF# 40-0393) for all the Ge–CPS. With the content of GeO_2_ increasing, the peak indexed to GeO_2_ (PDF# 83-0543) at 2 theta value of 25.96°, is detected, and its intensity gradually increases. At the same time, in the Raman spectrum (Figure 1d), the GeO_2_ has three vibrational peaks centered at 122.8 cm^−1^, 166.6 cm^−1^, and 442.6 cm^−1^, which can also be detected in the groups with higher GeO_2_ content.

### 3.2. Characterization of the CPS and Ge–CPS Bioceramics

The preparation process of the CPS and Ge–CPS bioceramics is presented in Figure 2a. As shown in Figure 2b, after sintering at 1200 °C, the main crystalline phase of CPS ceramic is still Ca_5_(PO_4_)_2_SiO_4_ (PDF# 40-0393). It can be seen in Figure 2c that after sintering at 1200 ℃, the main crystalline phase of GeO_2_ is still hexagonal GeO_2_ (PDF# 83-0543) while a broad peak at 2 theta angles of 25–35°, attributed to tetragonal GeO_2_ (PDF# 72-1149), can also be detected. From the XRD results of 0.15Ge–CPS in Figure 2d, the characteristic peak of CPS and the peak of hexagonal GeO_2_ (PDF# 83-0543) at 2 theta value of 25.96° remains. However, it should be noted that a broad peak at 2 theta angles of 25–35° was observed after sintering at 1200 °C, which indicates the existence of amorphous tetragonal GeO_2_ (PDF# 72-1149). As displayed in Figure 2e, it can be seen that many glass phases (red square) appear in the 0.15Ge–CPS ceramics. In addition, the element mappings of the 0.15Ge–CPS bioceramic are exhibited in Figure 2f and it can be seen that Ca, P, Si, and Ge are all evenly distributed across the surface of the ceramic. 

### 3.3. Release of Ge in PBS

The release of Ge in PBS was monitored within 28 days (as shown in Figure 3). It could be noticed that the release rate of Ge was fastest within 7 days of immersion. After that time point, the release became milder. However, all the Ge–CPS samples exhibited a sustained release of Ge. Moreover, the released Ge concentration increased with the increase in Ge content in CPS at each time point as expected. No Ge was detected in PBS in pure CPS specimens. The enlarged curves of the early stage releasing show that after 24 h, the released Ge concentrations for 0.05Ge–CPS, 0.10Ge–CPS, and 0.15Ge–CPS are 2.176 mg/L, 2.863 mg/L, and 3.497 mg/L, respectively. It can be seen that with the increase of GeO_2_ content in the Ge–CPS ceramics, the release of Ge increases.

### 3.4. Antibacterial Activity of Ge–CPS Bioceramics

The antimicrobial activities of Ge–CPS were evaluated based on bacterial adhesion analyses. In the results of the bacterial colony counting assay, it can be noted that for both bacteria cases, a mass of bacteria is distributed densely on the agar culture medium in the pure CPS group, while the number of bacteria decreases sharply with the increase of GeO_2_, in particular, very few *E. coli* colonies can be found in the 0.15Ge–CPS group, which demonstrates that Ge–CPS bioceramic scaffolds exhibit significant antibacterial activity against both *E. coli* and *S. aureus* (Figure 4a).

The morphologies of the bacteria cultured on the scaffolds for 1 day are presented in Figure 4b. On the pure CPS surface, plenty of *S. aureus* with sphere morphology is observed, which gather together as colonies and exhibit an even surface. In contrast, compared to those on the pure CPS surface, the *S. aureus* number declined and their morphology became irregular and smaller (the red arrow). In addition, it can also be noted that the membranes of *S. aureus* are destroyed. Furthermore, as is displayed in Figure 4c, the activity of *S. aureus* cultured with Ge–CPS samples dramatically decreases. Especially, it can be seen that few *S. aureus* can survive on the 0.15Ge–CPS scaffold surface (Figure 4b), and the proliferation rate of *S. aureus* cultured with the 0.15Ge–CPS scaffold is also the lowest, indicating that the 0.15Ge–CPS group has significant antimicrobial effect against *S. aureus*. Similar to the *S. aureus* case, the membranes of *E. coli* on the pure CPS sample exhibit intact morphology, while the bacteria number and viability of *E. coli* decrease dramatically on all three Ge–CPS surfaces. For both *S. aureus* and *E. coli* cases, the antibacterial activity of Ge–CPS shows a decrease with the increase in GeO_2_ content, although the drop in amplitude is not as big as that between CPS and Ge–CPS, indicating only 0.05 wt.% of GeO_2_ addition can endow CPS bioceramic with good antibacterial activity.

The antibacterial activity of Ge–CPS is further determined by the bacterial Live/Dead staining consequences. As displayed in Figure 5, large numbers of live *E. coli* and *S. aureus* dyed green can be detected on the CPS sample, while the dead bacteria which were dyed red can be barely found. On the surfaces of 0.10Ge–CPS and 0.15Ge–CPS scaffolds, the number of living bacteria decreases, while dead bacteria can hardly be found on these specimens either. On the 0.15Ge–CPS surface, both dead and live bacteria are dramatically reduced. The results show that a 0.15Ge–CPS sample can effectively restrain the propagation of *S. aureus* and *E. coli*. Moreover, the colonies of both *E. coli* and *S. aureus* on 0.15Ge–CPS specimens are dramatically less than those on other specimens. The corresponding inhibition ratio of the 0.15Ge–CPS sample against *S. aureus* and *E. coli* reached up to 72.1% and 76.4%, respectively (as shown in Figure 4c,d). It is worth noting that, as shown in Figure 4, the sample shows antibacterial effects even when the content of GeO_2_ is only 0.05. With the increase of GeO_2_ content, both the number and the viability of bacteria with Ge–CPS decreased, demonstrating that the growth of bacteria was greatly inhibited. The above results can be further confirmed by the bacteriostatic ring test in Appendix A. The CPS scaffold does not possess obvious inhibition diameters, while the inhibition area around the Ge–CPS scaffolds can be seen. These results show that the antibacterial property of Ge–CPS is related to the content of GeO_2_. Moreover, by combining the Ge-releasing curve (Figure 3), with the increase of Ge-releasing, the antibacterial property was also enhanced. 

### 3.5. Cytocompatibility of the Ge–CPS Bioceramics

In the study, rBMSCs are used to evaluate the cytocompatibility of Ge–CPS bioceramics. The proliferation results of rBMSCs are shown in Figure 6a. Overall, the intensity of cells cultured with all the groups, including pure CPS and three Ge–CPS bioceramic scaffolds, increases with the culture time, which indicates all samples have good cytocompatibility. After being cultured for 1 day, the proliferation of rBMSCs cells shows no significant difference among pure CPS and the three Ge–CPS samples. On Day 7, the cell viability of rBMSCs in Ge–CPS groups is even slightly higher than that in the pure CPS group. As shown in Figure 6b, the proliferation of rBMSCs above can be further verified through fluorescence images of nuclei and actin cytoskeleton staining. Clearly, after incubating for 24 h, rBMSCs demonstrate multipolar spindle morphology on all scaffolds. With the addition of GeO_2_, no obvious change is observed in either the cell number or morphology. Based on the results above, it can be concluded that an appropriate amount of GeO_2_ does not affect the cytocompatibility of CPS.

## 4. Discussion

The fabrication of antibacterial materials for special medical applications is of great significance to biocompatible implant materials. In the last decade, metal/ceramic composites have been widely explored to improve the antibacterial properties of implants [38,39]. As a white non-transition metal oxide, GeO_2_ has attracted considerable attention in the application of chemical catalysts, optical fibers, and semiconductors, due to its excellent chemical activity, superior electrical properties, and low cost [40]. Even though germanium is a nontoxic element noted for its antitumor, antiviral, anti-aging, and anti-inflammatory activities [41], its application in antibacterial biomedical implant materials has been scanty. This work was aimed to design a bone tissue implant material that is biodegradable, cytocompatible, and antibacterial. As an implant material, CPS is excellent in apatite formation, osteogenic activity, and biodegradability [42,43]. To investigate the antibacterial property of GeO_2_, a germanium-doped silicocarnotite scaffold that can locally provide Ge throughout the lifetime of the implant material was developed to reduce the risk of infection faced by patients. 

In this work, the Ge–CPS with designed Ge/(Ge + Si) molar ratios of 0%, 5%, 10%, and 15% were obtained. From the results of XRD and SEM images in Figure 2, besides the main crystalline phase of CPS and hexagonal GeO_2_, we can find the existence of amorphous tetragonal GeO_2_ (PDF# 72-1149) in 0.15Ge–CPS bioceramic. According to the literature, the melting point of GeO_2_ is 1115 °C, and under higher temperatures and pressure, phase transformation easily occurs [44]. When the sintering temperature increased to 1200 °C, the temperature was higher than its melting point part of the GeO_2_ melted and many glass phases occurred in Figure 2d. At the same time and a part of GeO_2_ was transformed from a hexagonal system to an amorphous tetragonal system which indicated that the GeO_2_ does not have an obvious reaction with the CPS matrix.

As we all know, the release curve is associated with the composition of the material and the immersion time [45,46]. In this work, with the increase of releasing time and Ge content in the material, the Ge concentration increases, which coincided well with the releasing trends of other elements reported in the literature [47,48].

Quantitative antibacterial testing showed that the Ge–CPS is antibacterial to both *E. coli* and *S. aureus*. Similarly in the antibacterial ring test, the CPS had no obvious effect on either bacteria strain, while Ge–CPS affected both strains. Besides, combining the Ge releasing curve, with the increase of Ge concentration the antibacterial efficacy was also enhanced. These results confirmed that the antibacterial property of Ge–CPS is related to the content of GeO_2_. It means that the antibacterial property is related to the amount of Ge released. From the picture of the live/dead staining pictures, in Figure 5, large numbers of live *E. coli* and *S. aureus* dyed green can be detected on the CPS sample, while the dead bacteria which were dyed red can barely be found. On the surfaces of 0.10Ge–CPS and 0.15Ge–CPS samples, the number of living bacteria decreases, while dead bacteria can hardly be found on these scaffolds either. What is more, both dead and live bacteria are dramatically reduced on the 0.15Ge–CPS surface. With the concentration of Ge increased, the live bacteria decreased, which was consistent with the results of the SEM images of the bacteria. The Ge–CPS can destroy the membranes of the bacteria which causes the death of the bacteria, and they cannot adhere to the surface of the materials [27], which means that the bacteria cannot survive on the surface of Ge–CPS.

However, the exact antimicrobial mechanism of Ge or GeO_2_ remains unknown. A lot of antibacterial agents can destroy the membrane of the bacteria by decomposing the polysaccharide [49,50]. In this study, as shown in Figure 4b, the membranes of the bacteria cultured on the surface of the Ge–CPS bioceramic scaffolds were destroyed. The morphology of the bacteria became irregular and their surface became coarse and uneven, which caused their death. Since they cannot adhere to the surface of the materials, the dead bacteria can hardly be found on the Ge–CPS groups in Figure 5. Based on the above results, it can be speculated that the released Ge interacted with the peptidoglycan on bacterial walls and caused subsequent damage to the bacterial membranes. However, the specific mechanism still needs to be investigated thoroughly by biological evaluation.

However, undue exposure to germanium has toxicological effects on the lungs [51] and kidneys [52]. To assess the toxicity of Ge–CPS, in this study, the cytocompatibility of Ge–CPS bioceramics was examined by testing the proliferation of rBMSCs. As proved in the literature, CPS is a material with excellent biocompatibility and osteogenic activity, it can promote the proliferation of rBMSCs [42]. In this work, as shown in Figure 6a, the Ge content of three Ge–CPS samples did not have an obvious effect on the proliferation of rBMSCs. It is known that the membrane of the cell is different from that of bacteria. For bacteria, the membrane contains cellulose, chitin, and peptidoglycan, while the human cell membrane contains phospholipids, protein, and carbohydrates [53,54]. Many anti-bacterial agents can destroy bacterial membranes by decomposing the polysaccharide which cannot kill normal cells at the same concentration [49,50]. Meanwhile, limited evidence suggested that HA, supplemented with germanium, may have a positive effect on bone regeneration relative to pure HA particles and other pure calcium phosphate phases [55]. As displayed in Figure 6b, rBMSCs demonstrate a multipolar spindle morphology on all scaffolds with all Ge–CPS bioceramics showing good cytocompatibility to osteogenic cells.

Based on the results, we believe that Ge–CPS possesses the following advantages. (1) GeO_2_ is distributed successfully in Ge–CPS and Ge is released slowly and steadily with the degradation of the material. (2) Ge–CPS demonstrated excellent antibacterial properties and can provide germanium in situ during the lifetime of the Ge–CPS bioceramic, thus preventing the patients from the risk of long-term infection. (3) Because of the good cytocompatibility, Ge–CPS bioceramic is an ideal implant material. Therefore, we believe that Ge–CPS has great potential in the treatment of infected bone defects in clinical settings.

## 5. Conclusions

Based on the significant antibacterial effect of GeO_2_, in the present work, Ge–CPS bioceramics were prepared by powder metallurgy method. Results demonstrated that the GeO_2_ does not change the main crystalline phase of CPS. Moreover, Ge–CPS can constantly release germanium elements. In addition, Ge–CPS can effectively inhibit the proliferation of *E. coli* and *S. aureus*, showing excellent antibacterial activity without sacrificing good cytocompatibility. Therefore, as a long-term implant material, Ge–CPS possesses a high potential for biological applications which could be a promising candidate for the repair of infected bone defects.

## Figures and Tables

**Figure 1 jfb-14-00154-f001:**
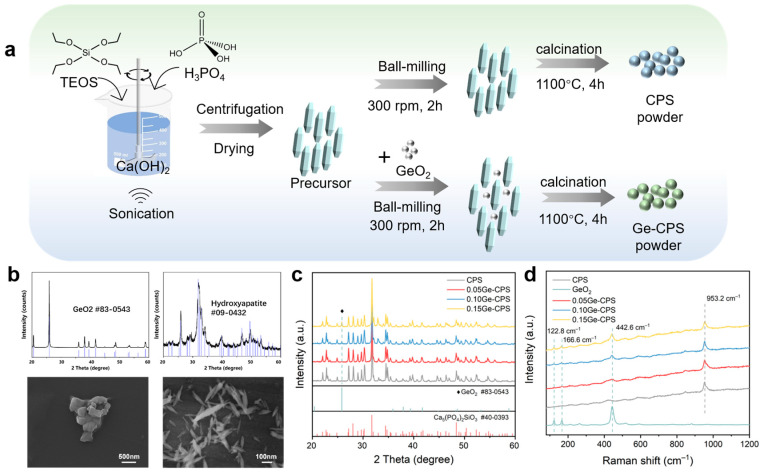
Synthesis and characterization of the calcined CPS and Ge–CPS powders. (**a**) A synthesis scheme of CPS and Ge–CPS powders by ultrasound-assisted aqueous precipitation method. (**b**) XRD patterns and SEM morphologies of the raw powders: CPS precursor and GeO_2_. (**c**) XRD patterns of calcined CPS and Ge–CPS powders. (**d**) Raman spectra of GeO_2_, calcined CPS, and Ge–CPS powders.

**Figure 2 jfb-14-00154-f002:**
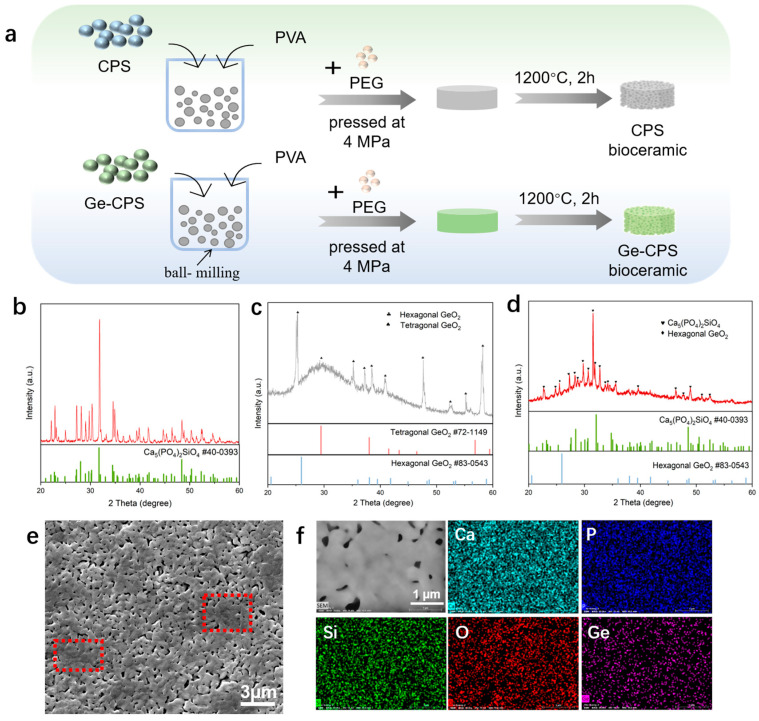
Synthesis and Characterization of the CPS and Ge–CPS bioceramics. (**a**) A preparation scheme of CPS and Ge–CPS bioceramics. XRD patterns of pure CPS ceramic (**b**), GeO_2_ (**c**), and 0.15Ge–CPS bioceramic sintered at 1200 °C (**d**). (**e**) SEM image of 0.15Ge–CPS bioceramic sintered at 1200 °C. (**f**) EDS mappings of 0.15Ge–CPS bioceramic sintered at 1200 °C.

**Figure 3 jfb-14-00154-f003:**
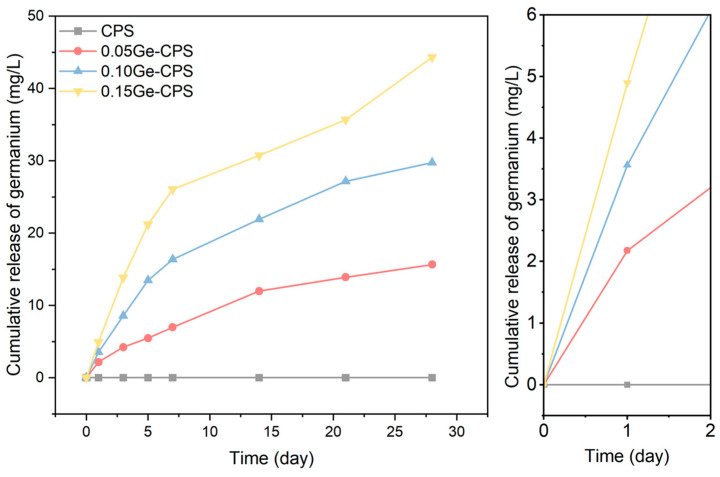
Cumulative release of Ge in PBS at 37 °C and enlarged curves of the early stage Ge releasing.

**Figure 4 jfb-14-00154-f004:**
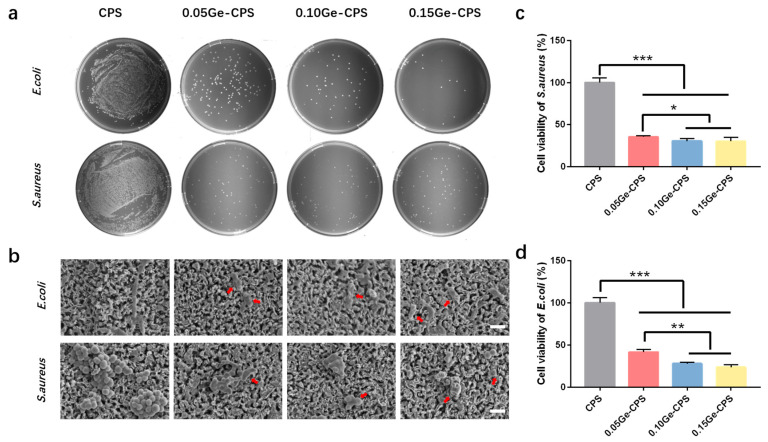
Antibacterial activity of the Ge–CPS scaffolds. (**a**) Photographs of re-cultured *E. coli* and *S. aureus* colonies on agar culture plates. (**b**) SEM morphologies of *S. aureus* and *E. coli* strains cultured on Ge–CPS scaffold surfaces for 24 h (scale bar = 2 μm); Cell viability of (**c**) *S. aureus* and (**d**) *E. coli.* * *p* < 0.05, ** *p* < 0.01, *** *p* < 0.001.

**Figure 5 jfb-14-00154-f005:**
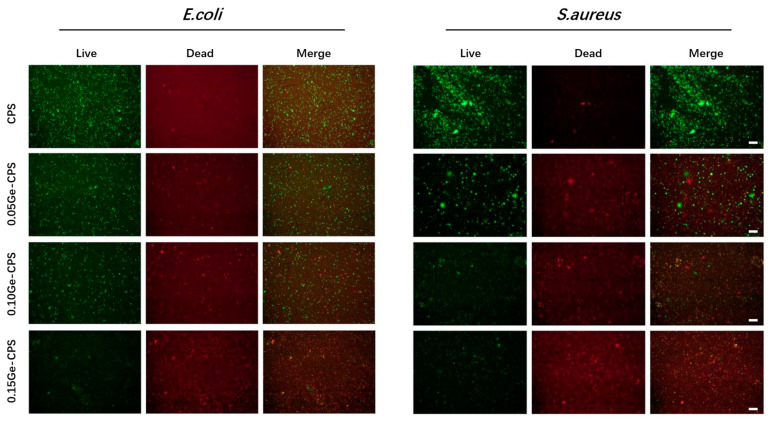
Live/dead staining results of *E. coli* and *S. aureus* (scale bar = 50 μm).

**Figure 6 jfb-14-00154-f006:**
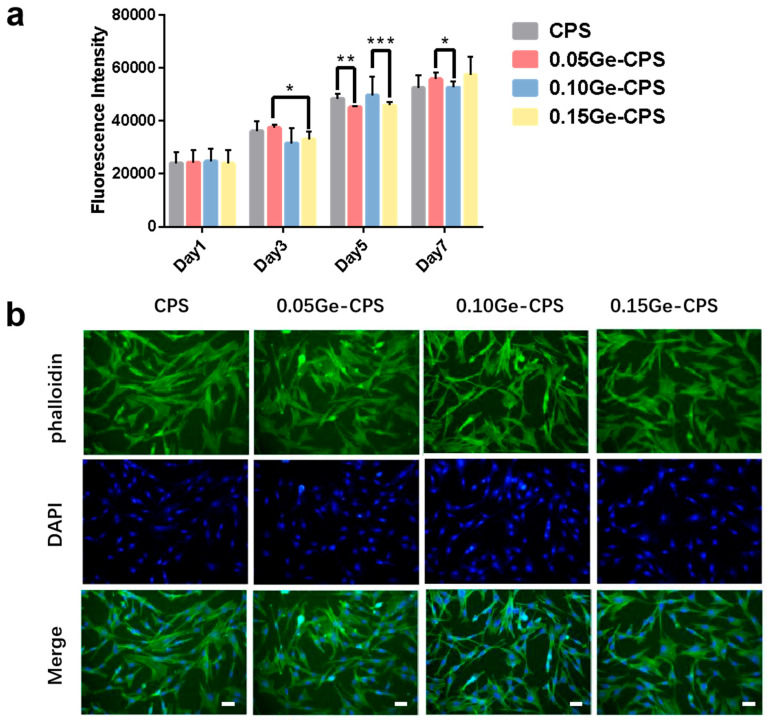
The cytocompatibility of the Ge–CPS scaffolds. (**a**) Fluorescence intensity of Alamar Blue produced by rBMSCs cells cultured with CPS, 0.05Ge–CPS, 0.10Ge–CPS, and 0.15Ge–CPS scaffolds. (**b**) Fluoroscopy images of F-actin stained with phalloidin (green) and nucleus stained with DAPI (blue) of rBMSCs cells cultured on CPS and Ge–CPS samples for 24 h (scale bar = 50 μm). * *p* < 0.05, ** *p* < 0.01, *** *p* < 0.001.

## Data Availability

The datasets generated during the current study are not publicly available, but are available from the corresponding author (cqning@shnu.edu.cn) upon reasonable request.

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
