# Peer review of "Realizing Both Antibacterial Activity and Cytocompatibility in Silicocarnotite Bioceramic via Germanium Incorporation"

_jfb, 2023, doi:10.3390/jfb14030154_

Round 1

Reviewer 1 Report

The present study investigates antibacterial properties of GeO2 as a component of bioceramic. The study design is very reasonable. All the neccessary tests are performed and the results are statiscically analysed. 

The text is very well written. I was not able to detect any major flaws. Some typos and minor grammatical mistakes were spotted here and there (see lines 17 and143, for example), but they do not impede the reading. 

I would advise the authors to slightly expand the discussion and address the state of GeO2 in the ceramics - especially rhe size of the particles. Am I correct in the understanding that GeO2 dous not react wwith the matrix and is present as separate particle? 

It is also interesting to discuss the mechanism of GeO2 antibacterial action. I do understand that this requires a separate study, but a brief section on this topic might be appropriate. For example, here is a study of GeO2 sols as nanozymes https://doi.org/10.1002/adfm.202001933 . Such sols seem to have antioxidant properties, while antibacterial agents (such as commonly used TiO2, ZnO, CuO ets) tend to be prooxidants. It might give some insight into the topic. 

Overall the suggestions above are merelly suggestions and the manuscript is sertainly fit for JFB without major modifications. 

Reviewer 2 Report

The manuscript provides information about the development of a bone tissue implant material cytocompatible, present antibacterial properties, and can release Ge continuously.

In my opinion, the manuscript is generally well structured, and the subject is interesting.

Although this, I have some issues/questions to address namely:

-Material section is missing also, suppliers should be added. Additionally, information such as molecular weight of % of purity is also missed;

-on eukaryotic and bacterial cell experiments the concentration of the particle tested is missing;

-on the particle characterization experiments regarding the dimension of the GE-CPs, polydispersity, as well as zeta potential, is missing as well as the proper discussion of these results and the comparison with literature;

- results of live/dead staining of the bacteria are too saturated. The laser intensity is too high to have proper results (this is particularly relevant to the dead analysis). Also, these live and dead results aren’t discussed. Why? Which concentration of particles was tested? The same amount applied to obtain the results in figure 2?

- I consider the discussion section very poor since:

                -the authors do not correlate the obtained results with the literature;

                - half of the discussion section is contextualization not the result from the discussion (I found this contextualization interesting…I just give the example to put it in proportion to the amount of information that you discuss- which was small);

                -you do not discuss or compared with literature any information about your particle characterization;

                -line 310-312- how do you conclude that if you performed experiments on 24h on bacteria also, analyzing the release of Ge after 24h it is the same for all the concentrations (at least is my perception after analyzing the release curve);

                - concerning the release curve subject- the release was what you expected? why? and literature? Could you compare it with other studies?

-If Ge kills bacteria why it did not have a cytotoxic effect? If the Ge releases are huge after 2/5 days and if Ge is toxic why do cells increase its proliferation after these time points? Which concentrations of your complexes did you test on bacterial experiments and cytocompatibility experiments? Was it the same concentration? If not…why?

                - concerning the cytocompatibility tests, why haven’t you a parallel between the results of the different Ge concentrations and the cytocompatibility?

Round 2

Reviewer 2 Report

I found the author's comments exciting and totally clarified my doubts. I want to ask them to add the information of the replies to points 8,9 and 10 since this information will enrich even more the manuscript.

Author Response

Responses to Reviewer 2 Comments

Dear Reviewer:

Thank you for your kind comments to “Germanium-doped silicocarnotite bioceramic with both antibacterial activity and cytocompatibility”. We revised the manuscript in accordance to your precious comments, and carefully proof-read the manuscript to minimize typographical, grammatical, and bibliographical errors. Revisions in the revised manuscript have been highlighted in yellow.

Detailed answers to the reviewer’s comments are as follows. 

Point 1: I found the author's comments exciting and totally clarified my doubts. I want to ask them to add the information of the replies to points 8,9 and 10 since this information will enrich even more the manuscript.

Response 1: Thank you so much for your advice, we revised the manuscript in accordance to your precious comments, and the information you mentioned above has been supplemented in line 369-372 and line 404-417.